# COVID-19-Associated Pulmonary Mucormycosis

**DOI:** 10.3390/jof8070711

**Published:** 2022-07-05

**Authors:** Vidya Krishna, Nitin Bansal, Jaymin Morjaria, Sundeep Kaul

**Affiliations:** 1Institute of Infectious Diseases, Apollo Hospitals, Chennai 600006, India; 2Division of Infectious Diseases, Rajiv Gandhi Cancer Institute, New Delhi 110085, India; nitin.bansal3011@gmail.com; 3Department of Respiratory Medicine, Royal Brompton and Harefield Hospital, Guy’s and St. Thomas Hospital NHS Foundation Trust, London UB9 6JH, UK; j.morjaria@rbht.nhs.uk; 4Departments of Respiratory Medicine and Intensive Care, Royal Brompton and Harefield Hospital, Guy’s and St. Thomas Hospital NHS Foundation Trust, London UB9 6JH, UK

**Keywords:** mucormycosis, fungal infections, respiratory infections, pulmonary mucor, COVID-19, SARS CoV-2, secondary infection, epidemiology, diagnostics, treatment

## Abstract

COVID-19-associated mucormycosis (CAM) emerged as an epidemic in certain parts of the world amidst the global COVID-19 pandemic. While rhino–orbital mucormycosis was well reported during the pandemic, in the absence of routine diagnostic facilities including lower airway sampling, pulmonary mucormycosis was probably under-recognized. In this review, we have focused on the epidemiology and management of COVID-19-associated pulmonary mucormycosis (CAPM). CAPM is a deadly disease and mortality can be as high as 80% in the absence of early clinical suspicion and treatment. While histopathological examination of tissue for angio-invasion and cultures have remained gold standard for diagnosis, there is an increasing interest in molecular and serological methods to facilitate diagnosis in critically ill patients and often, immune-suppressed hosts who cannot readily undergo invasive sampling. Combined medical and surgical treatment offers more promise than standalone medical therapy. Maintaining adequate glycemic control and prudent use of steroids which can be a double-edged sword in COVID-19 patients are the key preventative measures. We would like to emphasize the urgent need for the development and validation of reliable biomarkers and molecular diagnostics to facilitate early diagnosis.

## 1. Introduction

The COVID-19 pandemic which started in late 2019 took the world by surprise and is still with us today. Clinically, in addition to pneumonitis and multi-organ involvement, increased susceptibility to bacterial and fungal infections is well recognized [1]. Candida and Aspergillus infections are the most commonly cited COVID-19-associated fungal infections [2]. However, the under-reported mucormycosis carries a deadly prognosis. The unanticipated epidemic of invasive mucormycosis (IM) during the pandemic added extra layers of stress and complexity to the already overwhelmed healthcare systems in some countries such as India [3].

The interplay between viral respiratory pathogens and secondary fungal infections has been of considerable interest in the last few years, especially after the emergence of influenza-associated pulmonary aspergillosis (IAPA) during the 2009 H1N1 influenza pandemic [4]. Hundreds of cases of COVID-19-associated pulmonary aspergillosis (CAPA) have been reported from across the world [5,6]. COVID-19-associated mucormycosis gained attention much later in the pandemic when India, the diabetic capital of the world, reported thousands of cases. Rhino–orbital mucormycosis was the most commonly reported clinical presentation from India while few cases of disseminated, pulmonary and gastrointestinal mucormycosis were reported from the rest of the world [7]. It is highly probable that in the absence of routine diagnostic facilities and clinical guidelines, the exact burden of pulmonary mucormycosis was largely underestimated during the pandemic. Only recently, the Fungal Infection Study Forum and the Academy of Pulmonary Sciences of India have come out with a Delphi consensus statement on the definition, diagnosis and management of COVID-19-associated pulmonary mucormycosis (CAPM) [8].

In this review, we focus on the available literature on COVID-19-associated pulmonary mucormycosis (CAPM).

## 2. Aetio-Pathogenesis

Mucormycosis, a highly invasive fungal infection, is caused by a group of fungi under the group Mucorales [9]. The causative agents associated with mucormycosis include *Rhizopus* spp., *Mucor* spp., *Lictheimia* spp., *Rhizomucor* spp., *Cunnighamella* spp., *Apophysomyces* spp. and *Saksenaea* spp. [10]. It is frequently found on decaying organic matter and is present ubiquitously in tropical climates, especially in settings of high soil exposure. It enters the host through inhalation or inoculation. Rhino–orbital–cerebral, pulmonary, cutaneous and gastrointestinal diseases are the common clinical syndromes associated with mucormycosis. The disseminated form, which occurs due to hematogenous dissemination, is devastating.

Defects in the phagocytic action make the patient susceptible to invasive fungal diseases (IFD), including mucormycosis. Hence, disorders associated with neutropenia such as hematological malignancies and hematopoietic stem cell transplants, uncontrolled diabetes and use of systemic steroids are the standard risk factors of this invasive fungal infection [11]. Given the fact that the fungi are ubiquitous and there is a reasonable chance of exposure, when there is a defect in the immune system, the fungi can take an invasive route causing disease. It is hypothesized that hyperglycemia leads to the up-regulation of mammalian cell receptor (GRP78) that binds to Mucorales, enabling tissue penetration [12].

The use of systemic steroids, which is proven to be beneficial in managing COVID pneumonia, is one of the major reasons for the association of COVID-19 and mucormycosis [13]. However, this is not universal and there are CAM patients who were not exposed to corticosteroids leading to further hypotheses such as contamination of fomites, hospital air circulation system and oxygen supply chain and immune-paresis (lymphopenia) caused by the virus itself.

## 3. Magnitude of the Problem

As most of the COVID-associated mucormycosis (CAM) literature is restricted to retrospective data collection, the exact prevalence of the condition is difficult to estimate. A multi-center study in India quotes a CAM prevalence of 0.27% amongst hospitalized COVID-19 patients [14]. In this review, 187 cases of CAM were identified, out of which only 16 were pulmonary mucormycosis (PM) [14]. In a literature review from 18 countries, out of a total of 80 cases, only 20 were pulmonary [15]. Pal et al. reported 10 cases of PM out of a total of 99 cases of CAM [16]. Asian countries, particularly India, have contributed to the majority of these cases. Uncontrolled diabetes mellitus, COVID-19-related hypoxia and subsequent glucocorticoid use were found to be independent risk factors for CAM and associated with a very high mortality, ranging from 33% to 80% [14,15,16,17].

Rhino–orbital–cerebral mucormycosis constitutes the major clinical presentation in the literature of CAM. In global reviews on CAM, 50–80% of cases have been from India where rhino–orbital mucormycosis is easily identified due to the pre-existing association with diabetes mellitus [7,15]. However, the substantial difference in CAPM between India and rest of the world (7.3% vs. 21.4%) and the underrepresentation of non-diabetic hosts with COVID, e.g., transplant or oncology patients, imply that CAPM was either not suspected or reported in this large cohort [7]. CAPM was only diagnosed at autopsy in some cases and this is significant considering the paucity of these reports in the literature during the pandemic [18,19].

## 4. Clinical Features

Traditionally, PM is seen in hematological malignancies, where it typically presents as a pleuritic-type chest pain and hemoptysis. In fact, these two features have been proposed as differentiating features of PM from pulmonary aspergillosis, which is far more common in hematological malignancies, but they were not found to be very specific [20]. Overall, pulmonary mucormycosis can be very difficult to differentiate from other lung infections solely based on clinical features, but presence of pleuritic-type chest pain (accompanied by clinical signs of pleural effusion) may direct the physician towards PM.

Patients with PM can have a varied presentation ranging from fulminant acute pneumonia to an indolent cavity forming chronic pneumonia syndrome [21]. Presentation of COVID-19-associated PM is not different. A patient who has recently recovered from a moderate to severe COVID-19 (or worsened after initial improvement from COVID-19 pneumonia), with risk factors such as diabetes mellitus and hematological malignancies or other immunocompromising condition can present with fever, cough, chest pain and/or hemoptysis. In the multi-center Indian study, median time to diagnosis of mucormycosis after onset of COVID-19 illness was 18 days [14], and in another Indian study, median time was 19.9 days with more than 60% of patients being diagnosed with CAM after 8 days of onset of COVID-19 illness [7]; but in both of these studies, other forms of mucormycosis were also included. Due to the non-specific clinical–radiological features, it can be challenging to differentiate between worsening symptoms of COVID-19 pneumonitis, hospital-acquired bacterial co-infection or COVID-19-associated fungal infections, e.g., CAPA or CAPM. In critically ill patients, persistent fever, worsening inflammatory markers and sepsis with or without organ dysfunction unresponsive or partially responsive to antibiotics should trigger a search for invasive fungal infections, including pulmonary mucormycosis [18,22]. Further diagnostic work-up including radiology, airway sampling, histological sampling and advanced microbiological strategies may aid clinicians in reaching a diagnosis and treating appropriately. 

## 5. Diagnosis

Earlier diagnosis and initiation of appropriate anti-fungal therapy are key to achieving favorable outcomes in invasive mucormycosis (IM). However, there are a number of challenges, especially in COVID-19 patients, in establishing a diagnosis of pulmonary mucormycosis (PM), which include:(I).Lack of clinical suspicion: PM frequently remains unsuspected due to absence of specific clinical features and often being indistinguishable from severe pulmonary COVID-19. There are very few existing guidelines and clinical criteria that are COVID-19 specific and most are primarily suited for immunosuppressed hosts.(II).Challenges with imaging and appropriate sampling: Due to the risk of spread of the virus, there was considerable trepidation in performing aerosol-generating procedures during the pandemic. Additionally, there were issues due to patient transport pathways and elaborate cleaning protocols for radiology suites to obtain a computed tomography (CT) scan of the chest promptly.(III).Lack of reliable biomarkers: The fungal biomarkers beta-D glucan and galactomannan are poor markers of mucormycosis. This could be one of the reasons that mucormycosis is often less reported than aspergillosis in patients with viral respiratory infections like COVID-19 or influenza.

### 5.1. Role of Diagnostic Imaging

Diagnostic imaging is vital for early detection of invasive mucormycosis in patients with clinical suspicion. Certain distinguishing features of the lung parenchyma on a CT scan of the chest in patients with PM include multiple pulmonary nodules, often pleural-based, presence of lung cavitations and effusions (Figure 1, Figure 2, Figure 3 and Figure 4) [23]. The ‘reverse halo sign’ (central ground glass opacity surrounded by dense consolidation) is highly appreciated, mainly in neutropenic leukemic hosts. In a recent systematic review of CT findings in 16 patients with proven CAPM, consolidation and cavitation were seen in 11 (69%) patients, pleural effusion in 7 (47%), pneumothorax and nodules in 3 (19%) and ‘reverse halo sign’ and pulmonary embolism was seen in 2 (13%) patients [24]. Previously, Nam et al. reported that in hematologic patients, while consolidation and nodules are seen earlier in the disease process, central necrosis, cavitation and the air crescent sign are late findings [25]. This would suggest that CAPM is either under-diagnosed due to lack of differentiation from other fungal, including aspergillosis or bacterial pneumonias, or is diagnosed much later in the illness. 

Magnetic resonance imaging (MRI) is less useful than CT in the diagnosis of PM; however, it is more relevant for rhino–orbital–cerebral mucormycosis and it is generally helpful to identify soft tissue rather than bony abnormalities [26].

18F-Fluorodeoxyglucose (FDG) Positron Emission Tomography/Computed Tomography (PET/CT) combines functional, metabolic and structural modalities of imaging and can identify areas of infection and inflammation earlier and with more sensitivity than conventional CT scan. It works on the principle that areas of infection, inflammation or malignancy have increased FDG uptake as FDG acts as a glucose analogue and is taken up by the cellular glucose transporters. The local inflammatory response and increased vascular permeability also leads to higher uptake which is reported as maximum standardized uptake value (SUV max).

FDG PET/CT can be a particularly useful modality in patients with multiple risk factors for invasive fungal disease (IFD), to not only diagnose pulmonary fungal infections but also assess the extent of disease and follow up to assess residual infection status [27]. Recently, 18F- FDG–labeled leukocyte scan has been used to aid in the diagnosis of disseminated mucormycosis (pulmonary and intestinal) in a post-COVID-19 patient with other co-morbidities, including diabetes mellitus and post renal transplant status. The microbiological diagnosis was finally established by a terminal ileal biopsy using the imaging [28]. Hence, whole body imaging may also be helpful to identify alternate sites feasible for diagnostic biopsy.

### 5.2. Histopathology, Microscopy and Cultures

The demonstration of angio-invasiveness in lung tissue samples remains the corner stone in the diagnosis of PM as culture positivity rates are low. However, it is important to note that invasive tissue sampling is often delayed or not feasible due to the critical nature of the underlying illness.

Direct microscopy of bronchoalveolar lavage (BAL) specimens and staining with fluorescent brighteners such as Calcofluor White and Blankophor with examination under ultraviolet (UV) light helps demonstrate broad, aseptate hyphae suggestive of the Mucorales species (Figure 5) [23,29].

Lung tissue can be obtained by bronchoscopic guided trans-bronchial lung biopsy (TBLB), video-assisted thoracoscopic surgery (VATS) or open lung biopsy. While TBLB is the commonest diagnostic modality to obtain a tissue specimen, VATS and open lung biopsy are useful in patients undergoing surgical management.

On tissue specimens, Hematoxylin and Eosin (HE), Grocott Methanamine Stain (GMS) and Periodic Acid Schiff base (PAS) stains are used to demonstrate the fungal hyphae. Presence of broad (5–20 μm), irregular, ribbon like, aseptate or pauciseptate, wide angle branching hyphae are characteristic of Mucorales and should be differentiated from septate, narrow angle hyphae in aspergillosis. However, septations and angle of branching may be affected by tissue processing. Tissue samples may also show evidence of angioinvasion, necrosis, hemorrhagic infarction and neutrophilic infiltration (in non-neutropenic hosts). Chronic lesions may be pyogranulomatous with Splendore Hoeppli phenomenon (strongly eosinophilic amorphous material with radiating star-like or club-shaped-like configurations surrounding or adjacent to the fungal elements) [29]. PCRs on fresh or formalin fixed tissues and use of immunohistochemistry with monoclonal antibodies can be used to confirm the diagnosis. The techniques, though not widely validated or available, have increased specificity but with variable sensitivity [29].

Fungal cultures have only 40–50% positivity rates even when histopathology shows fungal hyphae and are usually positive by day 2 to 7 [29,30]. Matrix-assisted laser desorption/ionization time-of-flight (MALDI-TOF) mass spectrometry, though cumbersome, is likely to emerge as a promising tool for genus and species identification and has been evaluated by various authors [23,30]. 

Anti-fungal susceptibility testing is particularly useful in cases of treatment failure and also for epidemiological purposes. Lack of well-established clinical breakpoints remains a major limiting factor for routine testing [29,30]. Vitale et al. have studied in vitro susceptibilities of *Mucorales* sp. in 66 culture isolates and noted that amphotericin B was the most active drug, though somewhat less against *Rhizopus* and *Cunninghamella* species [31]. Posaconazole was the second most effective anti-fungal agent, though demonstrated reduced activity in *Mucor*, esp *M. circinelloides* and *Cunninghamella* strains. *Mucor* species were more resistant to azoles than *Rhizopus* species.

In a recent systematic review by Jeong et al., histopathology aided the diagnosis in 97%, cultures were positive in 78%. Genus and species-level identification was possible by morphological methods only in 53% and molecular methods were found to be promising [32].

### 5.3. Molecular Methods

Molecular detection of Mucorales DNA is generally used as an adjunctive method to histopathology, culture and microscopy for identification of Mucorales on culture isolates and tissue specimens. Fresh tissue specimens are preferred to formalin-fixed tissue specimens as more DNA can be extracted from the former [29]. The target genes include 18s rRNA, ITS or 28s rDNA regions [33]. PCRs assays can be pan-fungal, genus/species-specific multiplex PCRs or Mucorales-specific PCRs.

Mucorales DNA can be detected using PCR assays on clinical specimens including BAL samples, blood and urine. The commercially available pan-Mucorales real-time PCR assay MucorGenius^®^ (Pathonostics, Maastricht, The Netherlands) has been clinically evaluated in severely immunosuppressed patients with proven/probable PM as per EORTC-MSG criteria. This assay targets the 18s rDNA region of *Mucor* sp., *Rhizopus* sp., *Rhizomucor* sp. and *Cunnighamella* sp., without genus discrimination and had a sensitivity of 100% and specificity of 97.9%. They recorded a culture positivity rate of 3.1% compared to PCR positivity of 8.5%. MucorGenius^®^ PCR assay also picked up 10 possible cases of mucormycosis, all of which were culture negative [34].

While PCR-based methods can lead to rapid and early diagnosis, the issue lies with lack of standardization and clinical evaluation of assays [32]. Hence, their use is supported only with moderate strength as per the global European Confederation of Medical Mycology (ECMM) guideline [29].

### 5.4. Serological Methods


(i)Biomarkers: The current biomarkers for IFD like serum (1,3) beta-D-glucan assay (BDG) and serum and BAL Galactomannan do not detect mucormycosis as the cell wall of Mucorales species lack galactomannan and beta-D-glucan. (ii)There are no currently available serological tests for mucormycosis, though enzyme-linked immunosorbent assays, immunoblots and immunodiffusion tests have been evaluated with variable success previously [23,35].(iii)Mucorales-specific T cells: These have been studied in hematologic patients and need further validation in other populations. Potenza et al. have demonstrated the use of Mucorales-specific T cells as a surrogate diagnostic marker in hematologic patients in whom early an diagnostic biopsy is often challenging to perform [36]. In a small study involving 28 patients, they showed that Mucorales-specific T cells were detected only in patients with proven invasive mucormycosis at diagnosis and during the course of the disease. Importantly, they were neither detected before the onset of infection nor for long after resolution of the infection. Additionally, the Mucorales-specific T cells were not detected in patients with infections other than invasive mucormycosis or patients without infections. The effect of anti-fungal therapy on these T cells have not been studied.


## 6. Management

The common clinical and diagnostic features that lead to suspicion of pulmonary mucormycosis in COVID patients are listed in Table 1. While steps should be taken to confirm/refute the diagnosis, therapy should not be delayed as early initiation of appropriate therapy and combined medical and surgical management are important for improved outcomes. Optimal management also includes the reduction or discontinuation of immunosuppression including steroids and treatment of concomitant hyperglycemia and acidosis. 

### 6.1. Medical Management

CAPM is treated in keeping in lines with non-COVID-19 associated mucormycosis. The first line agent as per the European Confederation of Medical Mycology (ECMM) 2019 guideline for treating mucormycosis is liposomal amphotericin B, usually administered at a dosage of 5 mg/kg/day [29]. Doses up to 10 mg/kg/day may be used, especially if CNS penetration is required; though higher dosing is associated with greater increases in serum creatinine. Amphotericin B has moderate tissue penetration in lungs and levels are even lower in pulmonary epithelial lining fluid (ELF) compared to the lung tissue. No dosage adjustment is essential for liposomal amphotericin B in renal or hepatic dysfunction and in patients on continuous renal replacement therapy [37]. Amphotericin B lipid complex and amphotericin B deoxycholate are less preferred alternative agents. The latter is often the first line agent in developing countries due to lower costs, though is associated with significant nephrotoxicity with some increase in serum creatinine in 80% of patients and doubling of serum creatinine in 40% [29,37]. Other adverse effects more often seen with the deoxycholate preparation are infusion-related reactions, hypo- or hyperkalemia and anemia. 

Posaconazole and isavuconazole are the only effective azole agents active against mucormycosis. They can be used as less preferred alternatives to amphotericin B in first line therapy and also for salvage therapy [29]. The oral suspension of posaconazole is less preferred compared to sustained release tablets as it leads to lower drug exposure and needs to be administered with food for absorption, unlike the tablet formulation [38,39]. Co-administration of proton pump inhibitors decreases oral posaconazole bioavailability [40]. Intravenous (IV) posaconazole can be used as bridging therapy in critically ill patients until oral therapy can be commenced. Both the oral tablet and IV formulation of Posaconazole are dosed at 300 mg BD on day 1 followed by 300 mg once daily from day 2 of therapy (Table 2). While gastrointestinal side effects such as nausea, vomiting and diarrhea are the most frequent side effects, mild–moderate hepatotoxicity (1–3%) and prolongation of QTc interval (≤ 1% with tablets and IV formulation, 4% with suspension) are other notable side effects of posaconazole therapy. Hepatocellular failure, torsades de pointes and atrial fibrillation are a rarity. No dosing adjustment is required in hepatic or renal impairment for oral formulations, while the IV formulation is not recommended in moderate to severe renal impairment as the cyclodextrin component of the drug can accumulate leading to worsening renal impairment. Therapeutic drug monitoring is recommended in all patients with an aim to achieve a trough level of 1–3.75 mg/L whilst on therapeutic dosing [41]. A trough level of 1.8 mg/L is preferred in empirical therapy to cover all sensitive strains [42]. Studies show a 60% response rate (45% partial response, 15% complete response) for salvage therapy with posaconazole in patients with mucormycosis who were refractory to or intolerant of polyenes [43].

In a small single arm open label trial, isavuconazole has shown similar efficacy compared to amphotericin B for the primary treatment of mucormycosis [44]. Isavuconazole is available in both oral and intravenous formulations and is associated with lower hepatotoxicity and the shortening of QTc interval. Isavuconazole has predictable pharmacokinetics, excellent bioavailability and oral absorption is not affected by food intake. No dose adjustment is needed in renal impairment as there is no cyclodextrin component [45]. Both posaconazole and isavuconazole can have significant drug–drug interactions as they are both CYP3A4 inhibitors [39].

Table 2 lists the common dosing recommendations of polyenes and azoles used in the treatment of mucormycosis. Echinocandins and voriconazole have no activity against mucormycosis [37].

Treatment with amphotericin B is continued until clinical resolution and patients are then stepped down to oral posaconazole or isavuconazole, which is continued until radiological resolution and complete reversal of immunosuppression [29].

### 6.2. Combination Medical Therapy

Combination therapy with posaconazole or echinocandins (caspofungin) with lipid formulation amphotericin B backbone have shown some in vitro efficacy but there is poor clinical data to support routine use [46]. Combination with desferasirox and amphotericin B have failed to show benefit in a double-blind randomized control trial [47].

There is data emphasizing the combination of surgical debridement with systemic anti-fungal therapy leading to better outcomes as compared to anti-fungal therapy alone [48].

### 6.3. Surgical Management

Surgery should always be considered in patients with CAPM, though in some patients there might be issues with technical difficulties due to involvement of large vessels or mediastinal structures or in very sick patients with multiple co-morbidities. Surgery either in the form of resection, lobectomy or pneumonectomy improves survival when combined with appropriate medical therapy [48,49]. Surgical resection was a major factor associated with survival in multivariate analysis in a subgroup of patients with hematologic malignancies (odds ratio = 0.71) [50]. Chronic steroid intake or hypoproteinemia has been found to be associated with adverse surgical outcome as either a post-operative complication or in-hospital mortality [51]. While emergent surgery is warranted in those presenting with massive hemoptysis, in most cases, surgery should ideally be timed within 1–2 weeks of appropriate medical therapy, once metabolic abnormalities such as dysglycemia and dyselectrolytemias are corrected [8].

### 6.4. Prophylaxis

Apart from a small study which showed reduced incidence of CAPA in the prophylaxis arm but no survival benefit, no concrete evidence exists which investigates the use of anti-mold prophylaxis in COVID-19 ICUs. However, considering the emergence of multi-drug resistant fungal pathogens, nebulized formulations of amphotericin B may be worth considering; this modality is used in the post-lung transplant setting and also recommended as an alternative to posaconazole in high risk haematology–oncology patients [52].

## 7. Prognosis

Mortality is high in the absence of early initiation of anti-fungal therapy with surgical management and reported in various studies ranging from 73–88% [53]. The pooled mortality in CAPM (71%) is higher than the pooled mortality before the COVID-19 pandemic (57%). Delayed diagnosis, lack of appropriate treatment and lack of surgical intervention contribute to the increased mortality in CAPM. During the CAPM epidemic in India, there were severe shortages of amphotericin B and azole anti-fungals and this could have been a contributing factor. The presence of pulmonary artery pseudoaneurysm could be a significant cause for mortality in CAPM [54].

## 8. Prevention

Maintaining strict glycemic control and adhering to a protocol-based use of systemic steroids has been shown to prevent COVD-19-associated mucormycosis [55]. Anti-microbials and immunomodulators should be judiciously used [56]. Prolonged use of cloth (>4 h) or surgical masks (>6 h) has been found to be associated with CAM [57].

High environmental spore count and contaminated humidifier water and oxygen supplies may have contributed to the high disease burden of CAM in India [7,56,58].

Implementing strict protocols for sterilization and disinfection of shared equipment, ensuring infection control requirements are adequately maintained during construction activities, proper maintenance of ventilation systems and surveillance are important to prevent hospital outbreaks [58].

## 9. Conclusions/Future Prospects

CAPM is a well-recognized, lethal yet treatable complication of COVID-19 pneumonitis. Overall, there is a paucity of published data on CAPM and one important recommendation is to encourage the recording and reporting of cases via international registries (e.g., Fungiscope) who need to play an important part in increasing awareness as well as co-ordinating and recruiting for research studies. The potential for learning from each other cannot be over emphasized. 

Clinicians need to develop a high index of suspicion for CAPM and promoting enhanced surveillance for co-infections, including Mucor, as a part of routine clinical practice is essential.

Early diagnosis leading to appropriate and timely intervention is crucial for determining the overall outcome of patients with CAPM. The absence of a biomarker makes early diagnosis very challenging and yet is an important goal if we are to alter the prognosis of this deadly disease. This is an area where collaborative working is required between industry and clinicians to ensure appropriately designed studies are conducted leading to the provision of prognostic-altering diagnostic and therapeutic solutions. 

A combined co-ordinated approach can do much to improve patient outcomes from this under recognized devastating complication of COVID-19.

## Figures and Tables

**Figure 1 jof-08-00711-f001:**
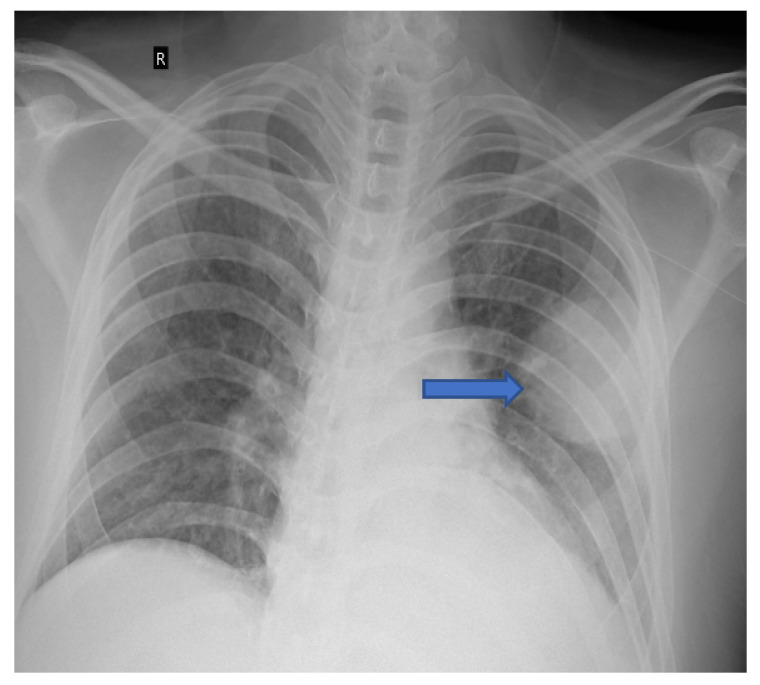
Chest radiograph (AP view) of patient showing round opacity in left lung (blue arrow).

**Figure 2 jof-08-00711-f002:**
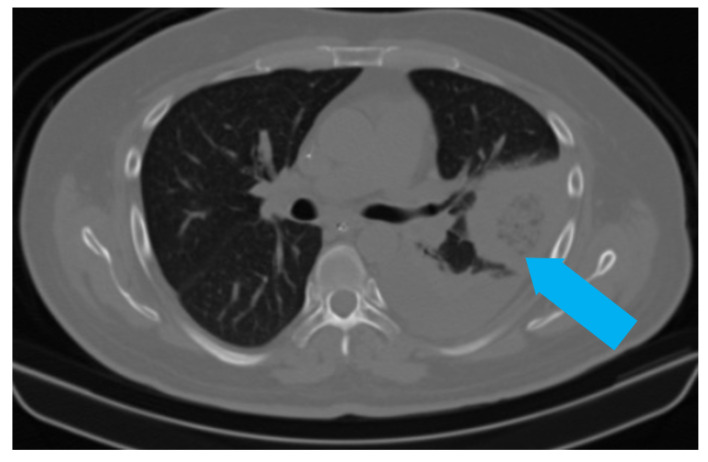
CT scan of the same patient showing a pleural-based consolidation with central clearing (blue arrow) and surrounding, denser, consolidation typical of reverse halo sign; also seen in left mild pleural effusion.

**Figure 3 jof-08-00711-f003:**
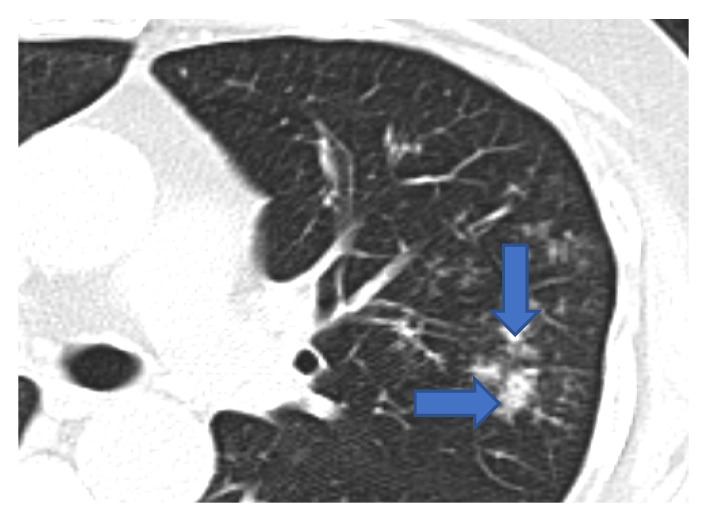
Multiple small nodules (blue arrows). Few show ‘tree in bud pattern’ in left lung parenchyma.

**Figure 4 jof-08-00711-f004:**
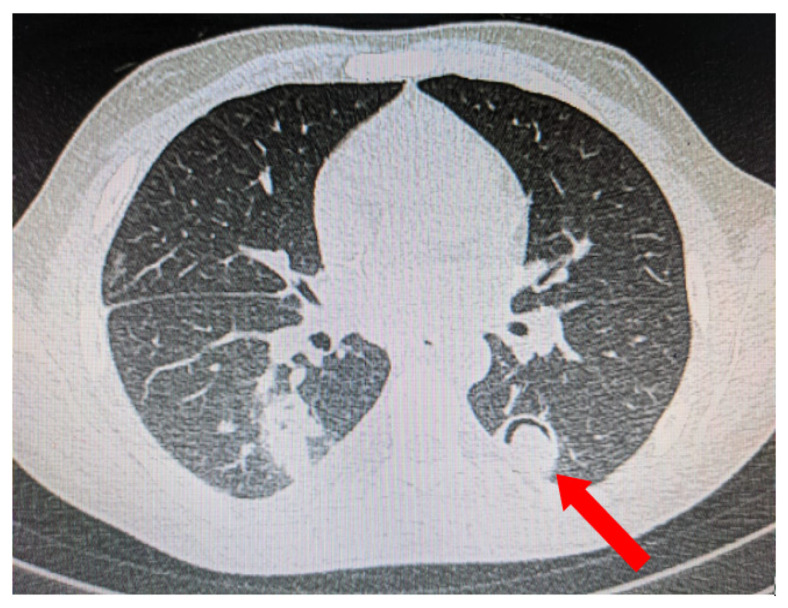
Pleural-based consolidation with cavity and a radio-opaque shadow within the cavity leading to formation of ‘air crescent’ (red arrow).

**Figure 5 jof-08-00711-f005:**
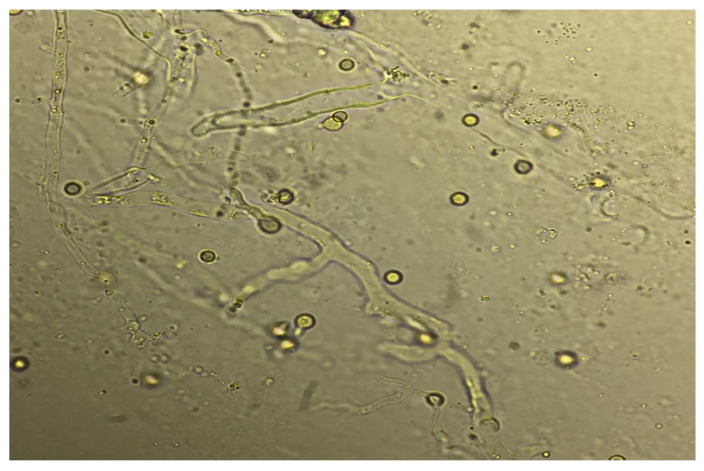
Broad aseptate hyphae indicating the Mucorales species seen on KOH stain of bronchoalveolar lavage (BAL) fluid.

**Table 1 jof-08-00711-t001:** Features that may guide clinicians towards COVID-19-associated pulmonary mucormycosis (CAPM).

Host Factors
1. Uncontrolled diabetes 2. Concomitant sinus, orbital involvement 3. COVID managed in mucor endemic areas such as India 4. Presence of pleuritic-type chest pain and hemoptysis
Radiological Features
1. Lung consolidation and/or cavity 2. Reverse Halo sign 3. Multiple lung nodules 4. Pleural effusion
Mycological Features
1. Negative BAL and/or serum galactomannan

The presence of these features does not rule out other mold infections.

**Table 2 jof-08-00711-t002:** Dosing recommendations of various anti-fungal drugs used to treat Mucormycosis in routine and special situations.

Drug	Preference	Dose	Remarks
Lipid formulations of Amphotericin B	First line agent	5 mg/kg; for obese patients (>100 kgs)—a fixed dose(i.e., cap at 500 mg rather than 5 mg/kg).	For renal impairment: Amphotericin B lipid complex should be avoided in patients with renal impairment because of its nephrotoxicity, other lipid formulations of amphotericin B can be given in standard doses.
Isavuconazole	Alternative first agent, step down or salvage therapy	Loading dose of 372 mg of isavuconium sulphate IV or oral q8h for 6 doses and then 372 mg of isavuconium sulphate IV or oral once a day. For pediatrics: 10 mg/kg of isavuconium sulphate three times a day for 2 days and then once a day.	No dose adjustment is required for renal impairment or mild to moderate hepatic impairment (CHILD-PUGH A and B) 50% dose reduction for severe liver dysfunction (CHILD-PUGH C). Avoid in presence of CNS disease as no data. Therapeutic drug monitoring (TDM) is not routinely recommended.
Posaconazole	Step down or salvage therapy	Delayed release tablets: 300 mg BD for 2 doses and then 300 mg once a day (for prophylaxis and therapy). Oral suspension: 200 mg thrice a day (for prophylaxis) and 400 mg twice a day (for therapy) IV formulations: 300 mg BD for 2 doses and then 300 mg once a day For pediatrics: Oral suspension: Oral suspension of posaconazole is recommended at 200 mg thrice a day for 6 months to 6 years and 300 mg thrice a day for 7–16 years. IV formulation: 6 mg/kg IV BD for 2 doses and then 6 mg/kg OD. Obese: Higher dose may be needed (TDM is highly recommended in this situation).	No modification is needed for renal and hepatic impairment (for IV formulation; cyclodextrin may get accumulated if GFR < 50 mL/min). Has a role in CNS disease (but should be restricted only as step down agent). TDM is recommended (1–3.75 mg/L).
Amphotericin B de-oxycholate	Not recommended routinely due to toxicities.	1–1.5 mg/kg per day. Saline hydration, use of pre-emptive anti-histaminic and prolonged duration of infusion (16–24 h) are some of the methods used to prevent infusion related side effects and nephrotoxicity.	In some situations (resource limited setting or due to non-availability of other drugs) this formulation may be utilized. Not recommended to be given in patients with renal failure; however, small doses of 25–50 mg have been applied during intermittent haemodialysis three times per week. In case of life threatening fungal infections, even standard doses of amphotericin B deoxycholate can be given with dialysis considering risk vs. benefit ratio.

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
