# Peer review of "COVID-19-Associated Pulmonary Mucormycosis"

_jof, 2022, doi:10.3390/jof8070711_

Round 1

Reviewer 1 Report

This is a manuscript reviewing COVID associated pulmonary mucormycosis. Although it is generally well written, it does not focus on CAM. There is a lot of information about diagnosis and treatment which is general and is not specific for CAM. These sections, especially regarding treatment, can be much shorter.

Please see some small comments in the attached file.

Author Response

Abstract
Line 27-28: You mention the importance of maintaining an international registry. Are you

aware of such a registry? Do you mean www.zygomyco.net or Fungiscope perhaps? Aetio-pathogenesis

Response: Yes. We have added this to the conclusion.

Line 68: I believe that you should not write “more” devastating. Something is or is not devastating.

Response: We have corrected this.

Line 76: mucorales should be written with a capital M (Mucorales). Furthermore, in other places in the manuscript it is written as Μurorales. This is not correct. Italics should be used only for genera and species.

Response: We have corrected throughout the manuscript.

Clinical features

Lines 105-108: Chest pain and hemoptysis are not specific for PM. Reference no.19 has very few cases, while in reference no.20, by Chamilos et al., chest pain and hemoptysis were not identified as differentiating features in multivariate analysis. Please correct accordingly.

Response: We have made the necessary changes.

Line 118: “critically patients” should be “critically ill patients”

Response: This stands corrected.

Diagnosis
Line 154: You wrote “either” but there is no “or” further down. Please rewrite this sentence. Line 194: “Calcofluour” should be “Calcofluor”

Response: The correction has been made.

Line 252: Please change to EORTC-MSG

Response: The correction has been made.

Management

Table 1: Host factors 1 and 2 are for rhino-cerebral mucormycosis not for PM. Pleuritic chest pain and hemoptysis are reasons to search for a fungal infection, but it can be mucormycosis, aspergillosis or even fusariosis.

Response: We appreciate the reviewer's comment and concur. However, we have clearly stated in the title of the table that these features 'may' help guide clinicians and are not pathognomonic for PM. We have made no amendments to the text of table 1.

Prognosis

The numbers you mention are from studies before the pandemic. Are there any newer data?

Response: Have added newer data instead of old data.

Prevention

Lines 399-400: The healthcare associated outbreaks with the risk factors you mention were mostly cutaneous mucormycosis, not pulmonary.

Response: We have removed the reference.

Conclusions
The last paragraph should be changed. It is not about mucormycosis.

Response: We have re-written the conclusions.

Reviewer 2 Report

1. The abstract doesn't reflect the content of article and not clearly structured. Abstract should consist of objectives, method, results and conclusion.

2. keyword too much

3. The discussion about clinical features and diagnosis is not sharp

4. Figures 1 and c can be added to the sign you want to explain.

5. In the description drawn D wrote a red arrow indicating the cavity leading to formation, but the red mark is not in the picture.

6. Is this image a self-portrait? if not then the literature citation must be given.

7. the discussion on surgical management, prognosis, and prevention is not in-depth

Author Response

  1. The abstract doesn't reflect the content of article and not clearly structured. Abstract should consist of objectives, method, results and conclusion.

         Response: This is a non-structured abstract in keeping with a review article.

  1. keyword too much

Response: We have limited the keywords to under 10 as per journal instructions.

  1. The discussion about clinical features and diagnosis is not sharp

           Response: We have modified these parts.

  1. Figures 1 and c can be added to the sign you want to explain.

          Response: We have added the signs.

  1. In the description drawn D wrote a red arrow indicating the cavity leading to formation, but the red mark is not in the picture.

       Response: this has been added.

  1. Is this image a self-portrait? if not then the literature citation must be given.

      Response: The images are from author’s own patient collections.

  1. The discussion on surgical management, prognosis, and prevention is not in-depth

    Response: We have corrected these parts further.

Reviewer 3 Report

The Manuscript ID: jof-1782918 entitled “COVID-19-associated Pulmonary Mucormycosis

is a very interesting review describing the main aspects of the pulmonary mucormycosis in patients with COVID-19. The manuscript is well written and of high interest of clinicians and mycologists around the word. In this reviewer opinion, it would be interesting if the manuscript could take the bellow few comments before be published in JoF Journal.

Specific comments:

1- Line 46: Please, include the CAPA reference from Salmanton-García et al. (Emerg Infect Dis. 2021;27(4):1077-1086).

2- Line 62-63: Please, double check the excess of space here. 

3- Line 90: The reference from Hoenigl et al. (number 14 in this manuscript) is already published in Lancet Microbe, 2022.

4- Line 168: There is no “red arrow” showing the formation of “air crescent” in Fig. D. Please, double check it. 

5- Lines 233, 255 and Table 2: There is a typo in “esp”, “100`%”, and “isavuconium sulphate IV or oral q8h for 6 doses”, respectively. 

Author Response

Specific comments:

  • Line 46: Please, include the CAPA reference from Salmanton-García et al. (Emerg Infect Dis. 2021;27(4):1077-1086).

Response: We have added the reference.

  • Line 62-63: Please, double check the excess of space here. 

Response: We have checked the spacing.

  • Line 90: The reference from Hoenigl et al. (number 14 in this manuscript) is already published in Lancet Microbe, 2022.

Response: We have quoted the above reference now.

       4- Line 168: There is no “red arrow” showing the formation of “air crescent” in Fig. D. Please, double check it. 

             Response: We have added the red arrow.

  • Lines 233, 255 and Table 2: There is a typo in “esp”, “100`%”, and “isavuconium sulphate IV or oral q8h for 6 doses”, respectively. 

Response: We have made the necessary changes.

Reviewer 4 Report

The Authors review a new interesting topic, becoming an increasing deadly disease in the pandemic era.

Globally the paper is well written, and is sufficiently clear. However the diagnostic paragraph is more detailed as compared to others, and too long, for example molecular and serological methods could be semplified.

Some more detailed observations:

Clinical features: the paragraph needs one sentence to discuss timing of occurrence of mucormycosis  in relation to COVID-19. considering the cases reported in the literature.

Diagnosis: figure D the legend indicates a red arrow, which is not present in the figure.

Line 290: please the sentence is not clear

Medical Management: the therapy is the same indicated for Mucormycosis, this should expressed at the beginning of the paragraph.

Posaconazole although the dosage is reported in the table, it should be cited in the text, as per liposomal Amphotericin. Moreover speaking about interaction of food PPI interaction reported in the literature should be cited.

The importance of surgery should be stressed.

Author Response

The Authors review a new interesting topic, becoming an increasing deadly disease in the pandemic era.

Globally the paper is well written, and is sufficiently clear. However the diagnostic paragraph is more detailed as compared to others, and too long, for example molecular and serological methods could be semplified.

Response: Thank you. We have simplified the diagnostics part.

Some more detailed observations:

Clinical features: the paragraph needs one sentence to discuss timing of occurrence of mucormycosis  in relation to COVID-19. considering the cases reported in the literature.

Response: We have added this to the manuscript.

Diagnosis: figure D the legend indicates a red arrow, which is not present in the figure.

Response: We have added the red arrow.

Line 290: please the sentence is not clear

Response: We have corrected it for clarity.

Medical Management: the therapy is the same indicated for Mucormycosis, this should expressed at the beginning of the paragraph.

Response: We have added this line.

Posaconazole although the dosage is reported in the table, it should be cited in the text, as per liposomal Amphotericin. Moreover speaking about interaction of food PPI interaction reported in the literature should be cited.

Response: We have added the dosing in the text and added the PPI interaction.

The importance of surgery should be stressed.

Response: We have corrected accordingly.

Round 2

Reviewer 2 Report

good revision